# Community factors and excess mortality in first wave of the COVID-19 pandemic in England

Bethan Davies ⓘ [1,2,3,4,7], Brandon L. Parkes[1,2,7], James Bennett[1,2,7], Daniela Fecht[1,2,7], Marta Blangiardo[1,2,3], Majid Ezzati ⓘ [1,2,5] & Paul Elliott ⓘ [1,2,3,4,6 ✉]

Risk factors for increased risk of death from COVID-19 have been identified, but less is known on characteristics that make communities resilient or vulnerable to the mortality impacts of the pandemic. We applied a two-stage Bayesian spatial model to quantify inequalities in excess mortality in people aged 40 years and older at the community level during the first wave of the pandemic in England, March-May 2020 compared with 2015–2019. Here we show that communities with an increased risk of excess mortality had a high density of care homes, and/or high proportion of residents on income support, living in overcrowded homes and/or with a non-white ethnicity. We found no association between population density or air pollution and excess mortality. Effective and timely public health and healthcare measures that target the communities at greatest risk are urgently needed to avoid further widening of inequalities in mortality patterns as the pandemic progresses.

[1] UK Small Area Health Statistics Unit, Department of Epidemiology and Biostatistics, School of Public Health, Imperial College London, Norfolk Place, London, UK. [2] MRC Centre for Environment and Health, Department of Epidemiology and Biostatistics, School of Public Health, Imperial College London, Norfolk Place, London, UK. [3] National Institute for Health Research Health Protection Research Unit in Chemical and Radiation Threats and Hazards, Department of Epidemiology and Biostatistics, School of Public Health, Imperial College London, Norfolk Place, London, UK. [4] National Institute for Health Research Health Protection Research Unit in Environmental Exposures and Health, Department of Epidemiology and Biostatistics, School of Public Health, Imperial College London, Norfolk Place, London, UK. [5] Abdul Latif Jameel Institute for Disease and Emergency Analytics, Imperial College London, London, UK. [6] UK Dementia Research Institute at Imperial College, Imperial College London, London, UK. [7] These authors contributed equally: Bethan Davies, Brandon L Parkes, James Bennett, Daniela Fecht. ✉email: p.elliott@imperial.ac.uk

During the first wave of the pandemic in early 2020, England experienced one of the highest death tolls from Coronavirus Disease 19 (COVID-19) in the industrialised world, beyond what would be expected from its underlying health status and factors like obesity[1–4]. Rates of diagnosed SARS-CoV-2 infections and deaths among people with confirmed infection varied substantially across England[5]. Geographic patterns in COVID-19 mortality have been reported in many settings and population density, urbanisation and air pollution are often mentioned as contributory factors in urbanised industrialised countries[6–9].

Excess mortality during the COVID-19 pandemic is the combination of deaths caused, or contributed to, by infection with SARS-CoV-2 plus deaths that resulted from the widespread behavioural, social and healthcare changes that accompanied national responses to the emergency[1,10–12]. Therefore excess mortality is a measure of the overall impact of COVID-19 on the population[13]. The direct impact of the pandemic is disproportionately affecting the elderly, people with chronic health conditions, people from a minority ethnic background and people who live in more deprived areas[3]. These are the same groups of people who have the greatest healthcare needs, outside of the pandemic, and therefore most at risk from disruptions to health services[14].

Studies have reported around 50,000 excess deaths at all ages nationally (England or England and Wales)[1,15–17]. The UK Office for National Statistics looked at mortality involving COVID-19 at the community-level (Middle Super Output Areas, MSOA) over the first 6 months of the pandemic and reported that age, ethnicity, urbanicity, and deprivation only partly explained which areas had higher mortality than the national average[18]. Local variations in all-cause mortality associated with the pandemic, and their community determinants, remain poorly understood. Here, we analysed geocoded data on all-cause mortality at ages 40 years and over for 6791 local communities (MSOAs) in England to quantify local variations in excess mortality in the first wave of the pandemic, from 1 March to 31 May 2020, and to identify the community characteristics associated with these patterns.

## Results

From 1 March to 31 May 2020, 174,327 people at ages 40 years and over died in England, compared with a mean of 121,441 deaths in the same period in 2015–2019, equivalent to 52,886 excess deaths. Compared with 2015–2019, a greater proportion of the deaths in 2020 were in men, in care homes and a smaller proportion occurred in hospitals (Fig. 1).

Because the local communities are small (MSOAs median population 7985 in 2018, median area 3.04 km[2], Supplementary Table 1), we used a Bayesian spatial model to obtain stable estimates of excess death rates based on data for each community and those of its neighbours to reduce uncertainty (Methods). The spatial model included terms for potential community determinants of mortality: percent population on income support as a marker of area poverty; population density; percent who are non-white; and percent population living in overcrowded homes. We also included air pollution, namely annual average nitrogen dioxide ($NO_2$) and fine particulate matter ($PM_{2.5}$) and number of care homes per 1,000 population. Data sources and definitions of these variables are detailed in Methods. Each variable was divided into quintiles of the distribution to allow for non-linear relationships (Supplementary Table 2).

All but 334 communities in men and 808 in women had higher mortality in 2020 than expected based on prior years (Fig. 2) with a posterior probability of increased mortality of at least 90% in 3711 (54.6%) communities in men and 2694 (39.7%) in women.

Of these, mortality more than doubled in 588 (8.7%) and more than tripled in 13 (0.2%) communities in men and in 444 (6.5%) and 13 (0.2%) communities respectively in women (Supplementary Table 3).

The communities with an increase in mortality were spread across the country with the lowest increases in remote rural areas (Fig. 3). The largest increases in mortality were concentrated in London, especially for men; for women high excess mortality also occurred in suburban areas. On average, communities with large increases in mortality tended to have greater social and environmental deprivation than those with small increases (Fig. 2).

The combination of a large relative increase in mortality and a high baseline (i.e. pre-pandemic) death rate meant that men in 2142 communities and women in 1527 communities experienced 250 or more excess deaths per 100,000 people aged 40 years and over compared with the prior years; in 336 communities for men and 326 for women, the excess mortality burden was at least 500 per 100,000 people. The large variation in excess death rates meant that 25% of all excess deaths during the pandemic occurred in only 9.0% of communities for men and 6.5% for women, and one-half of excess deaths occurred in 22.3 and 17.9% of communities, respectively. Excess deaths per 100,000 people were only moderately correlated between men and women (Supplementary Figs. 1 and 2).

Each of the community characteristics considered was individually (i.e. in univariate analysis) associated with excess deaths during the pandemic in graded fashion across quintiles (Fig. 4, Supplementary Table 4). However, there were strong intercorrelations between some variables; for example, Kendall's Tau was 0.67 and 0.55 between percent non-white population and levels of $NO_2$ and $PM_{2.5}$ respectively (Supplementary Table 5). When the community characteristics were considered jointly in multivariable analyses, air pollution and population density were no longer associated with excess deaths, (Supplementary Table 6).

Relationships with income support, percent non-white population and overcrowded homes persisted, although were attenuated – with a ~10% higher rate across quintiles for men, and somewhat weaker associations for women. The relationship with care home density, even after accounting for the other variables, remained strong, with a ~21% higher excess death rate for men and ~27% for women in communities with the highest compared to lowest density of care homes; many of these deaths were not assigned to COVID-19. Overall, the community variables accounted for 18.2% of the variation in mortality at community level in men and 15.3% in women (Supplementary Table 6). Local clustering explained a further 31.4% of variability in mortality for men and 19.2% for women, suggesting greater correlation in excess mortality between neighbouring areas for men than women. Sensitivity analyses with different smoothing parameters, excluding deaths in care homes, and combining data for men and women, did not materially alter our findings (Supplementary Table 7a–e)

## Discussion

In England, one of the worst affected industrialised countries in the first wave of the COVID-19 pandemic, we found substantial community-level variation in excess mortality, ranging from small declines to tripling in mortality in some areas. Although at first glance the high increases are more evident in cities, population density itself does not appear to be a driver of excess mortality on its own. Rather excess mortality risks are related to poverty, overcrowded homes, and non-white ethnicity, parallel to large impacts in communities where care homes are located; in England, many of these phenomena are more common in cities leading to the urban concentration of excess deaths. While we

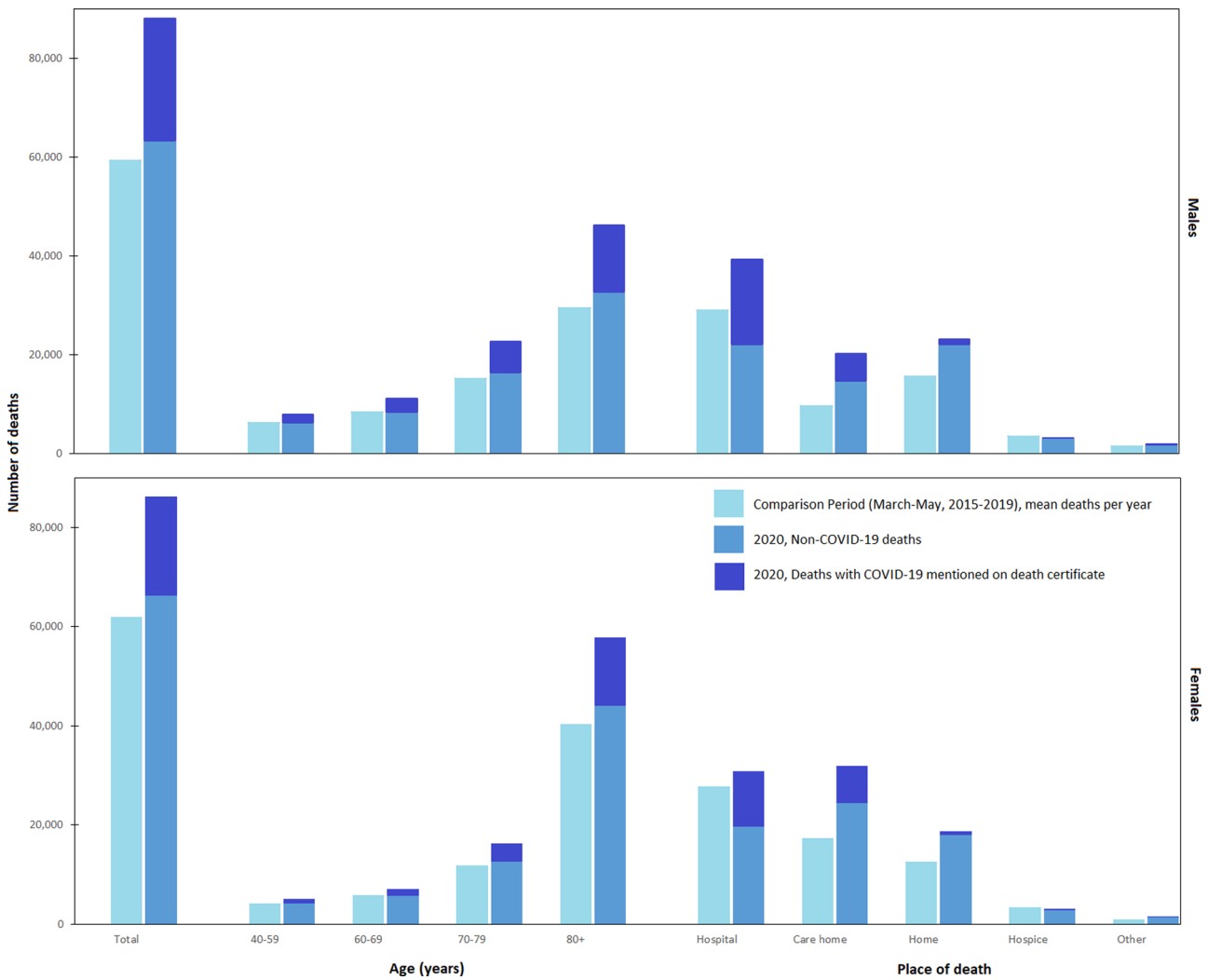

**Fig. 1 Distributions of deaths in England by sex, age and place of death.** Study period: first wave of Coronavirus Disease 19 (COVID-19) pandemic, 1 March to 31 May 2020. Comparison period: 1 March to 31 May 2015–2019.

found that these community factors and geographical clustering contributed independently to patterns of excess mortality, a large proportion of the variance remained unexplained. This underlines the importance of using real-time surveillance to identify local outbreaks, and their social and environmental patterns and determinants, and target public health resource according to need.

Our study has strengths and limitations. We included excess mortality from all causes, not just deaths coded to COVID-19. This gives a complete picture of the effects of the pandemic on mortality and is comparable across geographies, as it is not dependent on availability of testing or diagnostic facilities nor variations in coding practices. Not only could COVID-19 deaths have been wrongly ascribed to other causes but deaths from other causes may have been affected by the switching of healthcare resources to deal with the pandemic[2,14,19–21]. We used a Bayesian spatial framework to model excess mortality, incorporating random effects, in order to obtain age-adjusted and sex-specific stable estimates of excess deaths. We used a two-stage model, separately estimating the mortality rates for the comparison period (2015–2019) and then including these into the model for 2020, in order to fully propagate uncertainty from the comparison periods into the 2020 model, but without the latter influencing the rates for 2015–2019.

We did not directly account for the extent of spread of infection in the model. Rates of infection are associated with some community characteristics and adjusting for infection rates, in the second stage of the model, would likely detract from the true overall excess mortality associated with the community characteristics. Whilst accurate community-level data on COVID-19 incidence or seroprevalence during the study period are not available to formally test this assumption, a posterior comparison of antibody prevalence (antibody evidence of previous infection) at Lower Tier Local Authority geography ($n = 315$)[22] and excess mortality demonstrated a positive association (Spearman's Rank Correlation Coefficient 0.501, $p < 0.01$) (Supplementary Fig. 3).

Although we accounted for population change in the communities during the study period, population estimates at this scale were only available to 2019 and were extrapolated to 2020. We used mortality data for the same three months of the year (March to May) over the previous five years to estimate the expected numbers of deaths in those months during 2020. But factors like temperature may modify the number of deaths. In addition, we used data from the last national census in 2011 to obtain information on sociodemographic characteristics of communities. To the extent that there have been demographic changes in the nine years since then, this may have led to misclassification of areas with respect to their community characteristics.

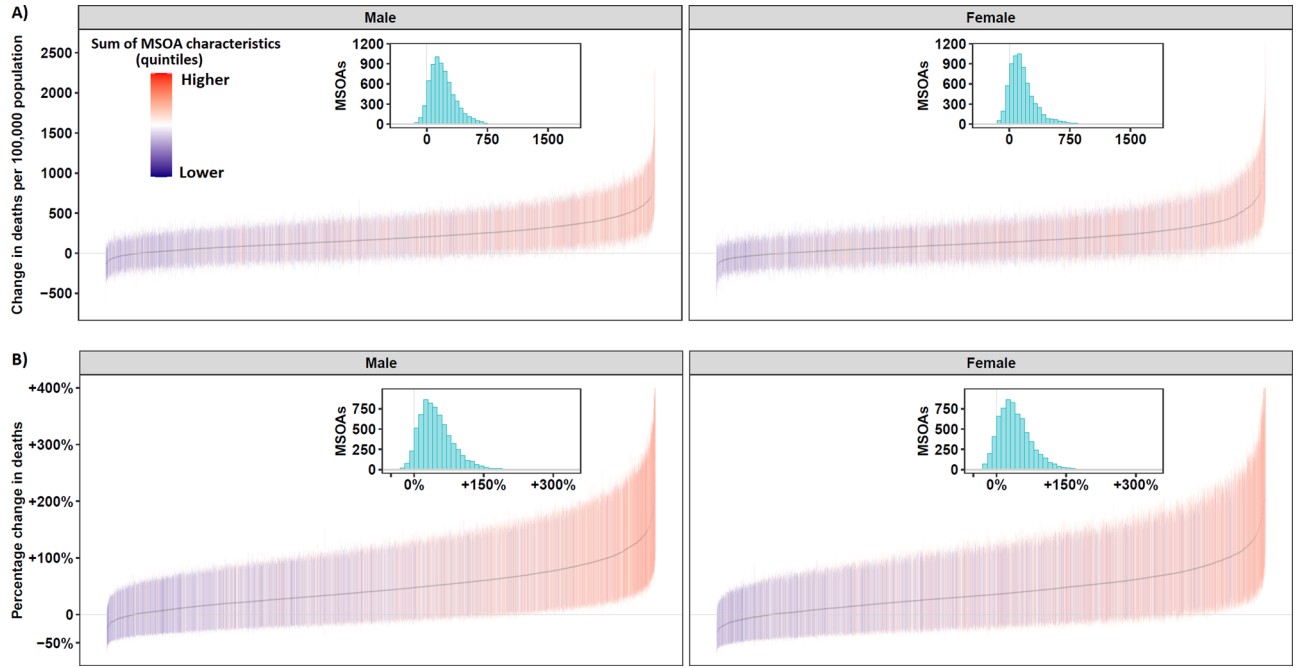

**Fig. 2 Change in mortality (unadjusted, with 95% credible intervals) in England from 1 March to 31 May 2020 compared to the same period for the preceding five years, for middle super output areas (MSOAs). A** Excess deaths per 100,000 people aged 40 years and over in 2020 compared to the average for the same period for the preceding five years ($n = 6{,}791$ MSOAs). **B** Percent increase in death rates in 2020 compared to the average for the same period for the preceding 5 years ($n = 6{,}791$ MSOAs). MSOAs ranked from lowest to highest excess mortality. The colour of the credible interval (from purple to orange) for an MSOA represents the sum ($n = 7$–$35$) of the quintiles it falls in for each of the seven community characteristics associated with excess mortality (% population on income support; population density; % population non-white; % population living in overcrowded homes; air pollution ($NO_2$ and $PM_{2.5}$); care homes per 1000 population). Inserts are histograms of the distribution of (**A**) excess deaths per 100,000 people aged 40 years and over, and (**B**) percent increase in death rates across MSOAs.

Our study provides important insights into the potential pathways leading to excess mortality during the pandemic. It includes a comprehensive set of community-level variables at the small-area level. Our estimate of 52,886 excess deaths in adults over 40 years of age in weeks 10–22 of 2020 is similar to other national estimates that range from 47,243 (all ages, England and Wales, weeks 11–19) to 57,300 (all ages, England and Wales, weeks 8–21)[1,15–17].

Our finding on the importance of care home density as a predictor of local excess mortality is consistent with the policy in the National Health Service to discharge up to 15,000 medically fit inpatients to avoid hospitals becoming overwhelmed[23]. It is likely that many of the elderly individuals discharged in this way will have needed support from social care services (including care homes) on discharge[24] and may not have been tested for the SARS-CoV-2 virus prior to discharge[25]. Many of these deaths (66% in men and 73% in women) were not attributed to COVID-19[26,27]. In addition, there was a higher proportion of deaths occurring in care homes during the first wave of the pandemic compared to 2019 which suggests that people may have had less access to hospitalisation at the peak of the epidemic[27].

Our study also underlines the associations between excess mortality and poverty, non-white ethnicity[28,29] and overcrowded housing[30,31] at the community-level. Population density and air pollution were not found to be associated to excess mortality, contrary to reports in some other studies where air pollution appeared to be a contributory factor for COVID-19 deaths[8,9,32,33]. This may reflect confounding since higher levels of air pollution are associated with deprivation and poverty that may be the drivers of the associations with COVID-19. Those living in poor communities may have fewer opportunities for adopting measures that reduce transmission, for example limiting travel[34] and working from home[35], have higher exposure to infection at work or may be more restricted in terms of accessing healthcare for COVID-19 and other conditions[36]. The index case within each household most likely acquired their infection within the community. But homes are the setting with the highest transmission rates of SARS-CoV-2[37,38], and avoiding close contact within the household may be particularly challenging in overcrowded premises. Recent and ongoing research indicates that higher risks associated with ethnicity may at least in part reflect higher levels of overcrowding and poverty (adjusted for in our analysis), higher representation in frontline jobs in the health and care sector[22,39], slower access to and utilisation of healthcare[30,36,40,41], and possibly higher rates of co-morbidities such as diabetes and obesity[3,42,43].

We have characterized community-level excess mortality in England and whilst we have identified patterns reported in other settings[44–47], our findings may reflect specific forms of structural inequalities present in England, so that generalisation outside this setting should be cautious. Further research to understand the pathways underpinning these associations is needed to inform long-term strategy to tackle the social and environmental drivers of inequality that may have contributed to differential mortality during the pandemic.

A major public health response from many governments has been either a national lockdown or a tiered lockdown applied primarily to cities[48]. Lockdowns in the first wave were highly effective at driving down the rates of new infection, but they are not sustainable[11,49–51]. Therefore, the immediate priorities are to bolster protection for care home residents and workers[52] and to continue to strengthen public health systems to ensure they have the capacity, in real-time, to test and diagnose newly infected individuals; identify their contacts; provide self-isolation and quarantine advice; and undertake national

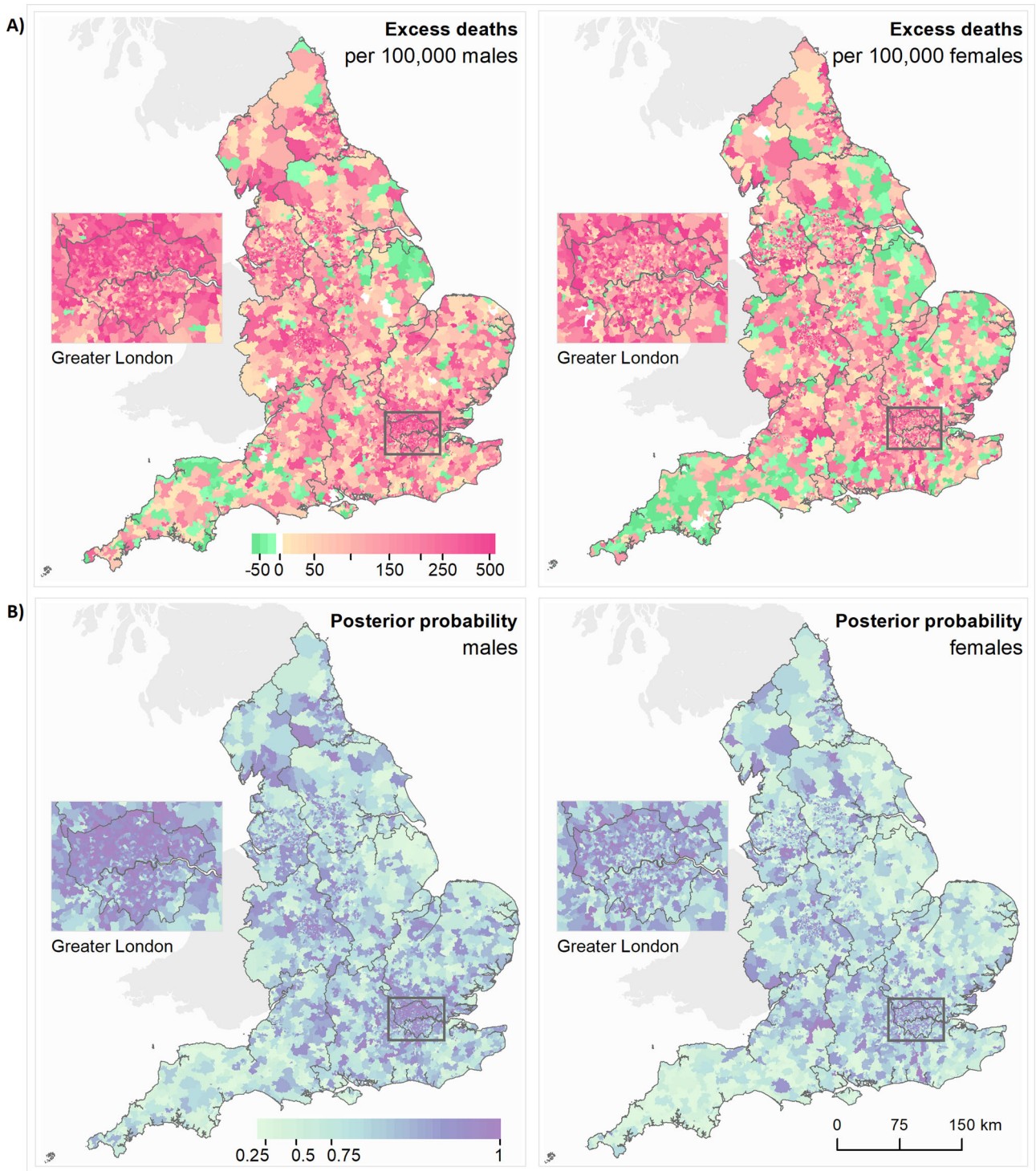

**Fig. 3 Maps of middle super output areas (MSOAs) in England showing excess deaths per 100,000 people aged 40 years and over. A** Excess deaths per 100,000 males (left)/females (right) from 1 March to 31 May 2020 compared to the same period for the preceding 5 years. **B** Posterior probability that excess deaths >0. Community characteristics of the MSOAs were: % population on income support; population density; % population non-white; % population living in overcrowded homes; air pollution ($NO_2$ and $PM_{2.5}$); care homes per 1000 population. We map the posterior probability that measures the extent to which an estimate of excess/fewer deaths is likely to be a true increase/decrease. Where the entire posterior distribution of estimated excess deaths for an MSOA is greater than zero, there is a posterior probability of ~1 of a true increase, and conversely where the entire posterior distribution is less than zero there is a posterior probability of ~0 of a true increase. This posterior probability would be ~0.5 in an MSOA in which an increase is statistically indistinguishable from a decrease (Supplementary Table 3). Contains OS data © Crown copyright (2020). Data available under the UK Open Government Licence v3.

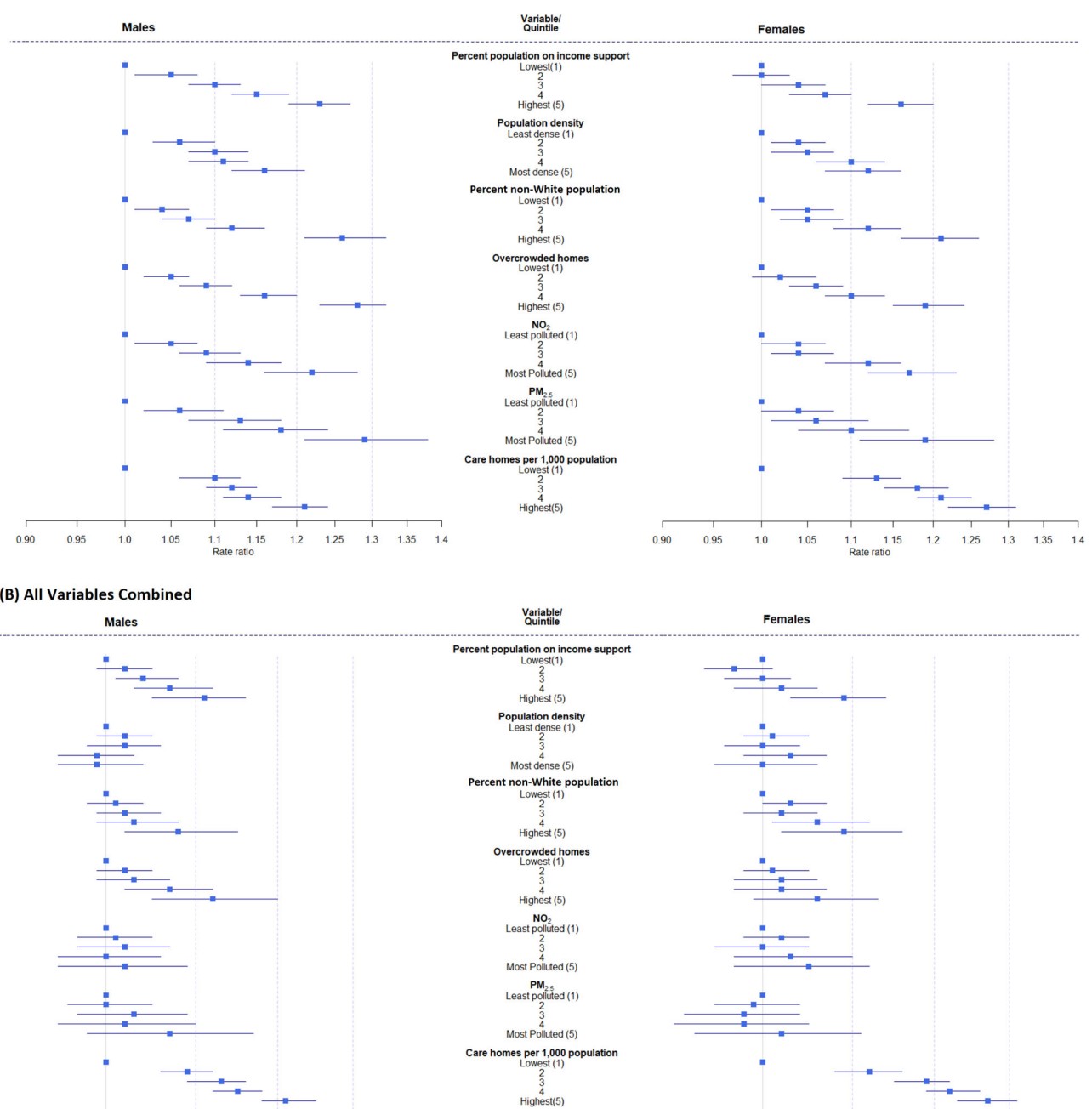

**Fig. 4 The relationship between community characteristics of middle super output areas (MSOAs) in England and excess mortality from 1 March to 31 May 2020 compared to the same period for the preceding five years.** Proportional increase in death rates shown as rate ratios (data are presented as posterior mean with 95% credible intervals) for quintiles of the distributions relative to lowest quintile. Males, n = 88,092 deaths in study period (1 March–31 May 2020), 296,985 deaths in comparison period (1 March–31 May, 2015–2019); females, n = 86,235 deaths in study period (1 March–31 May 2020), 310,220 deaths in comparison period (1 March–31 May, 2015–2019). **A** Univariable relationship between each characteristic and excess mortality, numerical values reported in Supplementary Table 4; **B** Multivariable relationship between characteristic and excess mortality after adjustment for the other characteristics, numerical values reported in Supplementary Table 6.

surveillance to inform the evolving policy response[53]. In parallel, economic interventions that support job security and provide financial compensation to low-paid workers required to self-isolate are essential to support population-level compliance with public health advice[54,55].

## Methods

**Data sources**. The mortality data used in this study were supplied by the Office for National Statistics (ONS), derived from the national mortality registrations and held by the UK Small Area Health Statistics Unit (SAHSU). We used data extracted by ONS on 14th January 2021. The ONS individual mortality data included date of death, date of registration of death, place of residence of the deceased, place of

death (e.g. hospital, hospice, care home, at home), and International Classification of Diseases tenth revision (ICD10) codes of the underlying cause of death. We limited analysis to people aged 40 years and over because the numbers of deaths in people under 40 years of age were small and would lead to unstable estimates in the small area analysis (COVID-19 age-specific death rate <14.3 per 100,000 in people aged under 40 years)[56]. Annual population was from ONS mid-year population estimates by age and sex for communities (MSOAs) in England, 2015 to 2019. No 2020 population data are yet available and 2019 estimates were used instead.

**Ethics and governance.** The study was covered by national research ethics approval from the London-South East Research Ethics Committee (Reference 17/LO/0846). Data access was covered by the Health Research Authority Confidentiality Advisory Group under section 251 of the National Health Service Act 2006 and the Health Service (Control of Patient Information) Regulations 2002 (Reference 20/CAG/0008).

**Characteristics of MSOAs (communities).** To investigate the association of community characteristics with excess mortality we included the following data at MSOA level:

- Income deprivation: Proportion of the population (adults and children, including asylum seekers) on government assistance due to low income and unemployment[57].
- Population density: Number of people per square kilometre from 2019 mid-year population estimates, as described above.
- Ethnicity: Percentage of the population of ethnic origin other than white including mixed ethnicities from 2011 census data[58].
- Housing: Percentage of overcrowded households defined as households with at least one fewer bedroom as required based on the number of household members and their relationship to each other, from 2011 census data[58].
- Air pollution: Annual average concentrations of nitrogen dioxide ($NO_2$) and fine particulate matter ($PM_{2.5}$) for 2018 at 1 km x 1 km grids, modelled to MSOA level using 2011 postcode headcount information[59].
- Location of care homes: Care homes per 1000 population using data from the Care Quality Commission via Geolytix[60].

All covariates were divided into quintiles, giving ~1360 MSOAs in each quintile.

**Statistical methods.** All analyses were carried out for males and females separately. We split age into four groups: 40–59 years; 60–69 years; 70–79 years; 80 + years.

We used a two-stage approach in order that the pandemic and comparison periods were treated as independent and distinct. First, we obtained estimates of the death rates in each MSOA for the comparison period of 1 March to 31 May 2015–2019 using a model that incorporated spatial and age terms to obtain stable estimates of death rates in each age group. Then in a second stage, we modelled the death rates from 1 March to 31 May 2020 (week 10 to 22), relative to the death rates estimated for the comparison period. We estimated excess mortality for each MSOA by comparing death rates for these three months between 2020 and 2015–2019 by sex and age-group. In the second stage, we included spatial and age terms as well as community variables to assess their effect on excess mortality. The spatial terms in both stages allowed for local smoothing across communities as well as global smoothing across England and were shared across all age groups.

In the first stage, we adjusted for age and smoothed over space to obtain stable estimates of the death rates for the comparison period. We assumed that the number of deaths $y_{itk}$ for the $i$th MSOA ($i = 1,…,6,791$), the $t$th year ($t = 2015,…,2019$) and $k$th age group ($k = 40–59, 60–69, 70–79, 80 +$) arose from a Poisson distribution:

$$y_{itk} \sim Poisson(\lambda_{ikt1}.Pop_{itk})$$

with the log-transformed death rates modelled as a sum of space, age and time terms:

$$\log(\lambda_{ikt1}) = \alpha_0 + \beta_{0k} + U_{0i} + V_{0i} + \gamma_t. \quad (1)$$

The common intercept for log-transformed death rates is represented by $\alpha_0$, with $\beta_{0k}$ the age effect for the $k$th age group. We modelled MSOA-level intercepts using a Besag, York and Mollie spatial model[61]; this includes spatially unstructured, independent and identically distributed Gaussian random effects ($V_{0i}$) and spatially structured random effects ($U_{0i}$) The latter were modelled with an intrinsic conditional autoregressive prior, which allows for death rates to be more similar across neighbouring MSOAs than those that are far away. This spatial model provides both local and global smoothing on the underlying death rate $\lambda_{ikt1}$.

We obtained the posterior distributions of the death rates in each MSOA, age group and year, $\lambda_{ikt1}$, and averaged over March to May for the 5 years of the comparison period (2015–2019) to obtain $\lambda_{ik1}$, the expected death rate for the $i$th MSOA and $k$th age group during March–May 2020 had there been an absence of the pandemic.

In the second stage, we estimated the ratio between death rates in March to May 2020 and the death rates we would have expected had there been no pandemic, using data for the same three months for 2015–2019. We estimated the effect of community variables on this ratio. For the number of deaths in the $i$th MSOA and

$k$th age group in 2020, we specified the following model:

$$y_{i.2020.k} \sim Poisson(\rho_{ik}.\lambda_{ik1}.Pop_{i.2020.k})$$

where $\rho_{ik}$ represents the age-specific ratio between death rates in 2020 and the comparison period ($\lambda_{ikt1}$).

We modelled the ratio $\rho_{ik}$ in a similar way to stage one using terms to account for both space and age:

$$\log(\rho_{ik}) = \alpha_1 + \beta_{1k} + U_{1i} + V_{1i}. \quad (2)$$

Community variables were incorporated into this second stage log-linear model to evaluate their effect on the mortality rate ratio. For univariable effects, we added the term $\delta X_i$ where $X_i$ is the quintile of the variable in the $i$th MSOA and $\delta$ is the associated effect. Similarly for the full multivariable model evaluating the joint effect of all variables we added $\sum_j \delta_j X_{ij}$ with $j=1,…7$.

To ensure uncertainty in the estimation of $\lambda_{ik1}$ in stage 1 is expressed in stage 2, we drew 200 samples from the posterior distribution of each $\lambda_{ik1}$ and ran a stage 2 analysis fixing $\lambda_{ik1}$ to each of these values in turn. For each of these 200 analyses, we sampled 200 values from the posterior distribution of each $\rho_{ik}$. In this way, we fully expressed the uncertainty resulting from the two stages of our analysis. The choice of the numbers of samples followed analyses using a range of different posterior distribution sample sizes. We ensured the stability of the estimates while minimising unnecessary computational burden. The final results presented in the paper are based on 200 samples from stage 1 and 200 samples from stage 2 (Supplementary Fig. 4 and Supplementary Tables 8 and 9).

For the neighbourhood variable effects, we report posterior mean and 95% credible intervals (2.5th to 97.5th percentiles) based on the 40,000 sampled values (200 × 200). In addition, for each MSOA we report both excess deaths per 100,000 people and the percentage change in deaths, as described below.

**Excess deaths per 100,000 people.** We obtained the posterior distribution of the estimated number of deaths across ages for March – May 2020, calculated as $\hat{y}_{i.2020\cdot} = \sum_k \lambda_{ik1} \times \rho_{ik} \times Pop_{i.2020.k}$, and subtracted the corresponding number for 2020 using the rates for 2015–2019, calculated as $\hat{y}_{i.2015-2019\cdot} = \sum_k \lambda_{ik1.2015-2019} \times Pop_{i.2020.k}$ where the samples for $\lambda_{ik1.2015-2019}$ are drawn in equal proportions from each of the years 2015–2019 allowing the uncertainty to be fully represented. This difference was then divided by the 2020 population over 40 years old in that MSOA and multiplied by 100,000. Figure 3A shows the excess deaths in map form: the colour key on the maps is categorical such that all MSOAs with excess deaths above 500 per 100,000 are coloured darkest red.

**Percentage change in deaths.** We obtained the posterior distribution of the estimated number of deaths across ages for March–May 2020 as summed over the age groups, as $\hat{y}_{i.2020\cdot} = \sum_k \lambda_{ik1} \times \rho_{ik} \times Pop_{i.2020.k}$ and divided by the corresponding number for 2020 using the rates for 2015–2019 as $\hat{y}_{i.2015-2019\cdot} = \sum_k \lambda_{ik1.2015-2019} \times Pop_{i.2020.k}$ where the samples for $\lambda_{ikl.2015-2019}$ are drawn in equal proportions from each of the years 2015 – 2019 allowing the uncertainty to be fully represented. We then subtracted 1 and multiplied by 100.

We fitted the models using the Integrated Nested Laplace Approximation (INLA), through the R-INLA software package (http://www.r-inla.org/)[62,63]. We specified a minimally informative prior, logGamma(1, 0.1), on the hyperparameters $\tau_V$ and $\tau_U$.

As sensitivity analyses, we re-ran the model using alternative priors for the hyperparameters $\tau_V$ and $\tau_U$ firstly using logGamma(0.5, 0.05), and secondly using the penalised complexity prior, as described by Moraga[64]. The results from these sensitivity analyses (Supplementary Table 7a, b) show little difference to the main model.

We carried out an analysis in which deaths in care homes were excluded (Supplementary Table 7c, d). Finally, we carried out an analysis in which deaths from both sexes were combined (Supplementary Table 7e).

**Reporting summary.** Further information on research design is available in the Nature Research Reporting Summary linked to this article.

## Data availability

- SAHSU does not have permission to supply data to third parties. No identifiable information will be shared with any other organisation. Individual mortality data can be requested through the Office for National Statistics (https://www.ons.gov.uk/).
- The results at MSOA level (excess deaths, credible intervals, posterior probabilities) used in Figs. 2 and 3 can be accessed at https://zenodo.org/record/4739256#.YJOvCC1Q0[65]
- Mid-year population estimates can be downloaded from https://www.ons.gov.uk/peoplepopulationandcommunity/populationandmigration/populationestimates/datasets/middlesuperoutputareamidyearpopulationestimates.
- English Index of Multiple Deprivation data can be downloaded from https://www.gov.uk/government/statistics/english-indices-of-deprivation-2019.
- 2011 Census data can be downloaded from https://www.ons.gov.uk/census/2011census/2011censusdata.

- Modelled air pollution data ($NO_2$ & $PM_{2.5}$) can be downloaded from https://uk-air.defra.gov.uk/data/pcm-data.
- Locations data of care homes can be downloaded from https://covid19.esriuk.com/datasets/e4ffa672880a4facaab717dea3cdc404_0.

## Code availability

The computer code written in R[66] for the two stages of Bayesian models used in this work is available on the GitHub repository[65].

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

## Acknowledgements

We thank Hima Daby, Gajanan Natu and Eric Johnson for their assistance in data acquisition, storage, preparation and governance; Aubrianna Zhu for her work on the online interactive map; and the Office for National Statistics (www.ons.gov.uk) for the provision of mortality data derived from the national mortality registrations. The work of the UK Small Area Health Statistics Unit is overseen by Public Health England (PHE) and funded by PHE as part of the MRC-PHE Centre for Environment and Health also supported by the UK Medical Research Council, Grant number: MR/L01341X/1), and the National Institute for Health Research (NIHR) through its Health Protection Units (HPRUs) at Imperial College London in Environmental Exposures and Health and in Chemical and Radiation Threats and Hazards, and through Health Data Research UK (HDR UK). P.E. is Director of the UK Small Area Health Statistics Unit. He acknowledges support from the Medical Research Council (MRC) for the MRC Centre for Environment and Health (MR/S019669/1); the British Heart Foundation Imperial College Centre for Research Excellence (RE/18/4/34215); NIHR Imperial College Biomedical Research Centre (BRC); the NIHR HPRU in Environmental Exposures and Health (NIHR-200880); and the NIHR HPRU in Chemical and Radiation Threats and Hazards (NIHR-200922). P.E. is supported by HDR UK and the UK Dementia Research Institute at Imperial College London, which receives funding from the UK Medical Research Council, Alzheimer's Society and Alzheimer's Research UK (MC_PC_17114). P.E. also acknowledges support from the Huo Family Foundation for research into COVID-19. J.B. and M.E are supported by Pathways to Equitable Healthy Cities grant from the Wellcome Trust (209376/Z/17/Z) and by a grant from the US Environmental Protection Agency (EPA), as part of the Centre for Clean Air Climate Solution (assistance agreement no. R835873). This article has not been formally reviewed by the EPA. B.D acknowledges funding from the NIHR HPRU in Chemical and Radiation Threats and Hazards (NIHR-200922) and the HDR UK Hub DISCOVER-NOW. All authors acknowledge infrastructure support for the Department of Epidemiology and Biostatistics provided by the NIHR Imperial BRC. The views expressed are those of the authors. This paper does not necessarily reflect the views of Public Health England, the National Institute for Health Research or the Department of Health and Social Care.

## Author contributions

B.D, P.E., and M.E conceived and supervised the study. B.D. and M.B. developed the initial study protocol. M.B. and J.B. developed the statistical model. D.F. prepared the population and covariate data. B.D. led the acquisition of mortality data and study permissions. B.P. authored the computer code, performed the analysis and prepared the initial results with input from J.B and M.B. All authors contributed to the drafting of the paper and critical interpretation of results and approve the final version for publication.

## Competing interests

The authors declare no competing interests.
