## [Peer Review File · Nature Communications]

REVIEWER COMMENTS

Reviewer #1 (Remarks to the Author):

This study relies on aggregated data on excess mortality and community characteristics for small statistical areas in England in order to analyze variation between areas in mortality during the COVID-19 pandemic. A key contribution is that authors rely on multivariate analyses to disentangle the independent effects on mortality variation of a set of often interrelated community covariates. There are some studies based on individual-level data on antecedents of COVID-19 mortality in Sweden that rely on similar covariates as those used here and that reach similar conclusions on the role of different variables in a multivariate study setup. I thus felt much reassured about the findings that were presented here. Another key contribution of the current study is its focus on community characteristics and the spatial dimensions of area-specific excess mortality during the pandemic. Some 18% of the variation in excess mortality between areas is ascribed to community characteristics depicted by the variables at authors' disposal, some 31% is ascribed to spatial correlation between areas. I think this is a key finding that could be better highlighted and its implications could be discussed in a more thorough way.

As already noted, I see no flaws in the basic study design but have a few suggestions on how to improve the presentation and clarify a few issues that was not entirely clear to me.

First, I would appreciate if the authors would add in the Title (and various Table and Figure headings) that the study is on England. Related to this, the authors could provide some discussion on the degree to which their findings may be specific to England or hold for a broader set of national contexts.

Many of the variables used stem from census data and it seems plausible that they relate to mortality variation both when measured at the individual level and when perceived as community characteristics. In the Swedish studies referred to above, it was found that marital status and the age composition of members within households were related to COVID-19 mortality. These factors may be less relevant in an area-based study design but the role of such factors in relation to, e.g., crowded housing could perhaps still be discussed.

The heading of Figure 2 appears somewhat mysterious for me. In Figure 3 I digest that we are presented patterns that are adjusted and un-adjusted for the covariates that are mentioned, but I don't see how the impact of the different covariates show up in the bars that are presented in Figure 2.

In the statements on data availability it is clear that individual-level mortality data are not publicly available. But the aggregated mortality statistics for all small areas could probably be made better visible.

Some more minor comments:

At page 2 authors seem to discuss health-care utilization in a way that appears to suggest that such utilization is a risk factor. These statements could be modified.

On p6 there is a discussion on the possible role of discharges of elderly people from hospitals to elderly care facilities. Perhaps there are also some patterns of denied admission to hospitals of people living in such facilities?

On p7 authors link their findings to a "now" situation; please specify the specific calendar period so that this can be read also in a future time dimension.

References

Drefahl, S., et al., 2020 A population-based cohort study of socio-demographic risk factors for COVID-19 deaths in Sweden. *Nature Communications* 11: 5097.

Brandén, M., et al., 2020. Residential context and COVID-19 mortality among adults aged 70 years and older in Stockholm: a population-based, observational study using individual-level data". The Lancet Healthy Longevity 1: e80-88.

Gunnar Andersson

Reviewer #2 (Remarks to the Author):

The manuscript applies a two-stage Bayesian spatial model to investigate inequalities in excess mortality at the community level during the first wave of the covid-19 pandemic in England. I enjoyed reading the manuscript and found it very interesting.

As described in the text, in the first stage the authors adjusted for age and smoothed over space to obtain estimates of the death rates for the comparison period. This first stage decomposes the log transformed rates as the sum of an overall intercept, an age specific intercept, a spatially structured random effect, an independent one, and a latent effect for the year, as the death rate estimates are based on data from 2015 until 2019. Then in the second stage they model the death rates for the same period in 2020 as a product among the death rate estimated in the first stage of the model, an age-specific ratio between death rates in 2020 and the comparison period, and the population size. As described in the text, community variables were incorporated into the second stage log-linear model to evaluate their effect on the mortality rate ratio.

While I find the proposed approach interesting, some of the hypotheses being made are not clearly explained in the text. Although the inference procedure is performed in two stages the authors try to account for the uncertainty in the estimation of the death rate between 2015-2019 by drawing 50 samples from the posterior distribution of the rate in each location and age group. Then they sampled 100 values from the posterior distribution of the adjusted rate for 2020. There is no justification for the choices of these sample sizes. Why did you not fit both models jointly? The reader would benefit from a discussion about this.

The two steps approach implicitly imposes an independent prior specification between the spatial and independent latent area effects ($U_{\{0i\}}$ and $U_{\{1i\}}$ and $V_{\{0i\}}$ and $V_{\{1i\}}$) in the modelling of the death rate and the age-specific ratio for the period 2015-2019 and age-specific ratio between death rates in 2020 and the comparison period, respectively. Why is this assumption reasonable? What is the motivation behind this assumption? Wouldn't be more reasonable to fit a joint model and allow for some prior correlation among these latent effects given that these components are capturing similar structures? Or is it the inclusion of covariates in the modelling of the age-specific ratio between death rates in 2020 that makes this prior independence a reasonable assumption?

The study does not find any association between population density or air pollution and excess mortality. Could this be due to some issue related to spatial confounding?

Overall the text is well written and easy to follow. I have one specific comment:

P. 15 l. 387 has λ_{ik2} been defined?

Reviewer #3 (Remarks to the Author):

The study conducted by Davies et al aimed to explore the excess mortality and its potential factors at community level during the first wave of the COVID-19 pandemic in England. A two-stage Bayesian spatial model was built to analyse the geocoded data on deaths in 2015-2020 and candidate covariates across over 6 thousand MSOAs in England for quantifying the local variations in excess deaths in the outbreak from March to May 2020. They found that multiple factors might

contribute to the inequality of excess mortality during the pandemic, but social and environmental variables only accounted for around 15% of the variation in excess mortality at community level, with no association found between excess mortality and population density or air pollution. Generally, the study is interesting and can improve our understanding on excess deaths and driving factors of inequalities across communities during the pandemic. Here are comments which could be addressed to improve the paper.

Major comments:

1. Is it a full research article? This article looks more like a letter or dispatch. I guess there are enough data/information to perform a thorough analysis, and describe, interpret and discuss results. Please expand it to a full-length article as required by Nature Communications.
2. Why did the model and covariates used in this study only explain a small proportion of the variation of excess mortality? The prevalence of COVID-19 must be highly correlated with excess deaths across space and time, but this study didn't include this factor. Why? Please clarify it.
3. Again, multiple publications have explored the factors associated COVID-19-related deaths in UK and other countries, and the underlying medical conditions, no. of hospital beds/ICUs/ventilators, the timeliness of treatment for COVID-19 and other diseases are also main drivers. This study should also incorporate these factors into the model if relevant data are available.
4. This study also found there were strong inter-correlations between some variables, and the multicollinearity might exist if these variables included together into the multivariable analyses, which might explain no association found between excess mortality and population density.
5. One of main limitations in this study is using the 2011 census data to define the community features as sociodemographic changes might be significant in the last 9 years. Additionally, as many countries including the UK are experiencing the 2nd wave, no sure how well these variations could explain the difference of excess mortality between waves.
6. It's unclear to me the predictive performance of the model to estimate the 'normal' mortality in 2020 without COVID-19 pandemic using historical data. It is the fundamental for estimating the excess mortality in this study. I'd suggest use the data from 2015 to 2018 to build the model and predict 2019 and compare with 2019 actual data to see how well it fits. Alternatively, the predictive performance can be examined by using January-February data in 2015-2019 to predict death rates in the same period in 2020.

Minor comments:

1. At first glance at the title, I thought it was a global study. I suggest adding the study country (i.e. England) in the title.
2. Line 48: the "effect" should be "affect"?
3. This study used the population estimates in 2018 for the analyses in 2019 and 2020, but similar datasets/estimates with population structures by year from 2010 to 2020 can be obtained from other data source, e.g. WorldPop (<https://www.worldpop.org/>), to improve the modelling in this research.
4. The adding value of this study compared with previous studies can be further interpreted and presented in the discussion.
5. The comparisons of estimates for excess deaths between this study and other publications (e.g. <https://jech.bmj.com/content/early/2020/10/15/jech-2020-214764>) can be discussed.

REVIEWER COMMENTS

Reviewer #1 (Remarks to the Author):

This study relies on aggregated data on excess mortality and community characteristics for small statistical areas in England in order to analyze variation between areas in mortality during the COVID-19 pandemic. A key contribution is that authors rely on multivariate analyses to disentangle the independent effects on mortality variation of a set of often interrelated community covariates. There are some studies based on individual-level data on antecedents of COVID-19 mortality in Sweden that rely on similar covariates as those used here and that reach similar conclusions on the role of different variables in a multivariate study setup. I thus felt much reassured about the findings that were presented here. Another key contribution of the current study is its focus on community characteristics and the spatial dimensions of area-specific excess mortality during the pandemic. Some 18% of the variation in excess mortality between areas is ascribed to community characteristics depicted by the variables at authors' disposal, some 31% is ascribed to spatial correlation between areas. I think this is a key finding that could be better highlighted and its implications could be discussed in a more thorough way.

As already noted, I see no flaws in the basic study design but have a few suggestions on how to improve the presentation and clarify a few issues that was not entirely clear to me.

R1.1. First, I would appreciate if the authors would add in the Title (and various Table and Figure headings) that the study is on England. Related to this, the authors could provide some discussion on the degree to which their findings may be specific to England or hold for a broader set of national contexts.

Authors' response:

We have included England in the title and figure 1 heading and consider the generalisability of the study (page 9, line 218).

R1.2. Many of the variables used stem from census data and it seems plausible that they relate to mortality variation both when measured at the individual level and when perceived as community characteristics. In the Swedish studies referred to above, it was found that marital status and the age composition of members within households were related to COVID-19 mortality. These factors may be less relevant in an area-based study design but the role of such factors in relation to, e.g., crowded housing could perhaps still be discussed.

Authors' response:

The data available to describe socio-economic status differs between England and Sweden, and as the Reviewer recognises, we used the data available in our setting. Information on overcrowding from the UK census is obtained at the individual level and then aggregated to the small-area before it is made available to researchers. We define the overcrowding variable in the methods (page 11, line 417).

We have expanded our discussion of the community characteristics identified in this study, including, adding a reference to a recent systematic review and meta-analysis by Thompson et al. and a

modelling study by Sun et al that identify households as the settings with the highest risk of transmission for SARS-CoV-2 (page 9, line 210).

R1.3. The heading of Figure 2 appears somewhat mysterious for me. In Figure 3 I digest that we are presented patterns that are adjusted and un-adjusted for the covariates that are mentioned, but I don't see how the impact of the different covariates show up in the bars that are presented in Figure 2.

Authors' response:

We have edited the figure legend for Figure 2.

R1.4. In the statements on data availability it is clear that individual-level mortality data are not publicly available. But the aggregated mortality statistics for all small areas could probably be made better visible.

Authors' response:

The data used in this study can be requested from the Office for National Statistics (ONS) and on publication, we will make our R code available on the GitHub repository (<https://github.com/smallAreaHealthStatisticsUnit>). We are bound by statistical disclosure control that prevents us from publishing cell counts of <3, with caution applied at <10 (mean deaths per MSOA is 25). ONS do not make these data publicly available by age or sex at geographies lower than Local Authority. We will provide a link to the results at MSOA level (excess deaths, credible intervals, posterior probabilities) used in figures 2 and 3.

Some more minor comments:

R1.5. At page 2 authors seem to discuss health-care utilization in a way that appears to suggest that such utilization is a risk factor. These statements could be modified.

Authors' response:

We have edited this sentence for clarity. Our intention is to highlight that the risk factors for the direct impact of COVID-19 (infection) and the indirect impacts (from disruption to healthcare) have been predicted to have the greatest impact on the same groups of individuals, thus widening the gap between their health status and that of the rest of the population.

R1.6. On p6 there is a discussion on the possible role of discharges of elderly people from hospitals to elderly care facilities. Perhaps there are also some patterns of denied admission to hospitals of people living in such facilities?

Authors' response:

We agree that rates of emergency department presentations and admission to hospital fell during the study period which we acknowledge on page 7 line 154. We have added a reference to the Office for National Statistics citing the increase in the proportion of deaths (for non-COVID-19 causes) occurring in care homes in 2020 compared to 2019 (page 8, line 196).

R1.7. On p7 authors link their findings to a “now” situation; please specify the specific calendar period so that this can be read also in a future time dimension.

Authors' response:

We have edited the paragraph to ensure it is contemporary to the current context.

References

Drefahl, S., et al., 2020 A population-based cohort study of socio-demographic risk factors for COVID-19 deaths in Sweden. Nature Communications 11: 5097.

Brandén, M., et al., 2020. Residential context and COVID-19 mortality among adults aged 70 years and older in Stockholm: a population-based, observational study using individual-level data”. The Lancet Healthy Longevity 1: e80-88.

Gunnar Andersson

Reviewer #2 (Remarks to the Author):

The manuscript applies a two-stage Bayesian spatial model to investigate inequalities in excess mortality at the community level during the first wave of the covid-19 pandemic in England. I enjoyed reading the manuscript and found it very interesting.

As described in the text, in the first stage the authors adjusted for age and smoothed over space to obtain estimates of the death rates for the comparison period. This first stage decomposes the log transformed rates as the sum of an overall intercept, an age specific intercept, a spatially structured random effect, an independent one, and a latent effect for the year, as the death rate estimates are based on data from 2015 until 2019. Then in the second stage they model the death rates for the same period in 2020 as a product among the death rate estimated in the first stage of the model, an age-specific ratio between death rates in 2020 and the comparison period, and the population size. As described in the text, community variables were incorporated into the second stage log-linear model to evaluate their effect on the mortality rate ratio.

While I find the proposed approach interesting, some of the hypotheses being made are not clearly explained in the text. Although the inference procedure is performed in two stages the authors try to account for the uncertainty in the estimation of the death rate between 2015-2019 by drawing 50 samples from the posterior distribution of the rate in each location and age group. Then they sampled 100 values from the posterior distribution of the adjusted rate for 2020.

R2.1 There is no justification for the choices of these sample sizes.

Authors' response:

In general the sample size needs to be large enough to ensure that it represents well the posterior distribution of the parameters to be estimated; in this study while increasing the sample size for stage 2 is not computationally intensive, a large sample size in stage 1 implies high computational burden as the stage 2 model needs to be run for every sampled value in stage 1. More generally large sample sizes might add little in terms of precision in the estimates. We have undertaken a post hoc analysis, presented in the Extended Data Figure 4, and Extended Data Tables 8 and 9, that describe how the MSOA estimates of excess mortality change for different sample sizes from the posterior distributions in stage1-stage2. These show that the mean absolute error between MSOA estimates from different sample sizes is small and the correlations very strong with no meaningful gain from increased sample sizes beyond the 50, 100 sample sizes originally used. Nevertheless, all results and figures now use the increased sample sizes of 200 at stage 1 and 200 at stage 2.

R2.2. Why did you not fit both models jointly? The reader would benefit from a discussion about this. The two steps approach implicitly imposes an independent prior specification between the spatial and independent latent area effects ($U_{\{0i\}}$ and $U_{\{1i\}}$ and $V_{\{0i\}}$ and $V_{\{1i\}}$) in the modelling of the death rate and the age-specific ratio for the period 2015-2019 and age-specific ratio between death rates in 2020 and the comparison period, respectively. Why is this assumption reasonable? What is the motivation behind this assumption? Wouldn't be more reasonable to fit a joint model and allow for some prior correlation among these latent effects given that these components are capturing similar structures? Or is it the inclusion of covariates in the modelling of the age-specific ratio between death rates in 2020 that makes this prior independence a reasonable assumption?

Authors' response:

The choice of a two-stage model has been motivated by the fact that, while we wanted uncertainty from the estimates of the rates for the comparison period (2015-2019) to be propagated to the estimates for 2020, we did not want the data from 2020 to influence the rate estimates for 2015-2019. Although this means that we are not explicitly modelling the spatial effects as correlated between 2015-2019 and 2020, our modelling approach allows for a degree of similarity since we include λ_{ik1} into the second stage model; we include its spatial structure modelled through U_{0i} and V_{0i} as well. We have added a paragraph to the discussion (page 7, line 156) .

R2.3. The study does not find any association between population density or air pollution and excess mortality. Could this be due to some issue related to spatial confounding?

Authors' response:

Population density and the air pollution variables were associated with excess mortality in the unadjusted analysis, but the effect became non-significant in the model with the complete set of community-level co-variates. We have added a sentence to the discussion (page 9, line 201). There were some intercorrelations between the variables included in the multivariable analysis (Kendall's Tau from -0.20 to 0.67; see Extended Data Table 5). However, in case of multicollinearity we would expect to see unstable estimates (with wide standard deviations) but this is not what was observed.

R2.4. Overall the text is well written and easy to follow. I have one specific comment:

P. 15 l. 387 has λ_{ik2} been defined?

Authors' response:

We now define it as ρ_{ik} .

Reviewer #3 (Remarks to the Author):

The study conducted by Davies et al aimed to explore the excess mortality and its potential factors at community level during the first wave of the COVID-19 pandemic in England. A two-stage Bayesian spatial model was built to analyse the geocoded data on deaths in 2015-2020 and candidate covariates across over 6 thousand MSOAs in England for quantifying the local variations in excess deaths in the outbreak from March to May 2020. They found that multiple factors might contribute to the inequality of excess mortality during the pandemic, but social and environmental variables only accounted for around 15% of the variation in excess mortality at community level, with no association found between excess mortality and population density or air pollution. Generally, the study is interesting and can improve our understanding on excess deaths and driving factors of inequalities across communities during the pandemic. Here are comments which could be addressed to improve the paper.

Major comments:

R3.1. Is it a full research article? This article looks more like a letter or dispatch. I guess there are enough data/information to perform a thorough analysis, and describe, interpret and discuss results. Please expand it to a full-length article as required by Nature Communications.

Authors' response:

We have updated the structure of the submission to format it as a full-length article.

R3.2. Why did the model and covariates used in this study only explain a small proportion of the variation of excess mortality? The prevalence of COVID-19 must be highly correlated with excess deaths across space and time, but this study didn't include this factor. Why? Please clarify it.

Authors' response:

It is not unusual in this type of study for covariates to explain a small proportion of variation. The variation not explained by community variables can be explained partly by stochastic variation, and other community factors not included in our analysis. Indeed COVID-19 prevalence is likely to explain some of the variation.

The model does not include COVID prevalence for the following reasons: (1) individuals were exposed to infection in part as a result of the community characteristics already in the model; in other words, infection is associated with the community characteristics and inclusion of sero-prevalence in the model would detract from the true overall excess mortality consequences of the community characteristics; (2) the lack of testing capacity in England during the early stage of the pandemic means that prevalence data (especially at small area level) is unreliable. (Testing was initially limited to hospitalized patients, gradually expanding to symptomatic key workers, and then symptomatic people in the general population by mid-May).

We have presented in a plot the relationship between the available seroprevalence data (which is at a lower spatial resolution, n=317; Extended Data Figure 3) and excess mortality and added a paragraph to the discussion describing this (page 7, line 164). In addition, we also show below the

relationships between seroprevalence and % of non-white population and % of population on income support.

Plot of % non-white population at LTLA level ($n = 317$) against antibody prevalence from REACT-2 study (round 1, June 20th – July 13th 2020) (Ward et al., 2020). Pearson's product moment correlation 0.676. When data organised into quintiles, Kendall's rank correlation tau 0.411

Plot of % of population on income support at LTLA level (n=317) against antibody prevalence from REACT-2 study (round 1, June 20th – July 13th 2020) (Ward et al., 2020). Pearson's product moment correlation 0.237. When data organised into quintiles, Kendall's rank correlation tau 0.138

R3.3. Again, multiple publications have explored the factors associated COVID-19-related deaths in UK and other countries, and the underlying medical conditions, no. of hospital beds/ICUs/ventilators, the timeliness of treatment for COVID-19 and other diseases are also main drivers. This study should also incorporate these factors into the model if relevant data are available.

Authors' response:

This study focuses on the community and environmental factors and excess mortality during the pandemic. We have excluded (where available), area-level data about the structure of the NHS (e.g. number of ICU beds) for the following reasons: (1) healthcare and co-morbidity prevalence are intermediate variables on the pathway from community-level exposures to mortality. See our response to R.2.3; (2) secondary and tertiary healthcare providers in England do not have catchment areas and there is transfer of patients between providers to match resources to need; (3) there was a marked increase in critical care capacity during the first wave of the pandemic (including surge capacity in seven Nightingale Hospitals). We have undertaken a post hoc analysis into the associations with sero-prevalence of antibodies to SARS-CoV-19 and excess mortality (see response to R3.2).

R3.4. This study also found there were strong inter-correlations between some variables, and the multicollinearity might exist if these variables included together into the multivariable analyses, which might explain no association found between excess mortality and population density.

Authors' response:

See the response to R2.3.

R3.5. One of main limitations in this study is using the 2011 census data to define the community features as sociodemographic changes might be significant in the last 9 years. Additionally, as many countries including the UK are experiencing the 2nd wave, no sure how well these variations could explain the difference of excess mortality between waves.

Authors' response:

The UK Census 2011 is the most comprehensive and complete source of demographic information in the UK. The next census will be conducted in 2021 but outcomes at the small area level are not expected to be released until beginning of 2022. Other potential data sources, for example, from surveys exist but this information is not available at the population level. We acknowledge the limitations of 2011 census data applied to the 2020 population of England in the discussion. It is however the highest quality national level data available for the included covariates.

R3.6. It's unclear to me the predictive performance of the model to estimate the 'normal' mortality in 2020 without COVID-19 pandemic using historical data. It is the fundamental for estimating the excess mortality in this study. I'd suggest use the data from 2015 to 2018 to build the model and predict 2019 and compare with 2019 actual data to see how well it fits. Alternatively, the predictive performance can be examined by using January-February data in 2015-2019 to predict death rates in the same period in 2020.

Authors' response:

We have now performed the analysis largely as suggested. For each of the years 2014 to 2019 we have used our model trained using the other 5 years in that period to predict the mortality (1 March to 31 May) and compared our model's predictions against the actual mortality across the 6,791 MSOAs. The results can be seen in the tables below. The consistently high coverage across sets of years suggests that our approach to estimating expected mortality is robust to year-to-year variation. The model successfully predicts the mortality in a particular year based on the training years; natural variability in mortality (driven by, for example, 'flu deaths or cold weather) is expressed in the uncertainty of the predictions.

Table R1. Predictive power of model of excess mortality.

A. Males

Year	Training years	Coverage* N (%)
2014	2015-2019	6,654 (98.0%)
2015	2014,2016-2019	6,741 (99.3%)
2016	2014,2015,2017-2019	6,756 (99.5%)
2017	2014-2016, 2018, 2019	6,764 (99.6%)
2018	2014-2017,2019	6,710 (98.8%)
2019	2014-2018	6,752 (99.4%)

B. Females

Year	Training years	Coverage* N (%)
2014	2015-2019	6,623(97.5%)
2015	2014,2016-2019	6,726 (99.0%)
2016	2014,2015,2017-2019	6,740 (99.2%)
2017	2014-2016, 2018, 2019	6,750 (99.4%)
2018	2014-2017,2019	6,723 (99.0%)
2019	2014-2018	6,737 (99.2%)

* Coverage is the number (%) of MSOAs where 95% credible interval of the modelled mortality encompasses the actual for that year.

Minor comments:

R3.1.1. At first glance at the title, I thought it was a global study. I suggest adding the study country (i.e. England) in the title.

Authors' response:

See our response to R1.1.

R3.1.2. Line 48: the "effect" should be "affect"?

Authors' response:

This has been changed in the manuscript.

R3.1.3. This study used the population estimates in 2018 for the analyses in 2019 and 2020, but similar datasets/estimates with population structures by year from 2010 to 2020 can be obtained from other data source, e.g. WorldPop (<https://www.worldpop.org/>), to improve the modelling in this research.

Authors' response:

Since initial submission of this work, mid-year 2019 population estimates have become available from the data source used in this study (ONS). We have updated all analysis using this population estimate data on 2019. In addition, we used a more recent extract of mortality data from ONS which contains additional death registrations (there is always some delay in the reporting of a very small minority of deaths, e.g. those referred to the coroner).

R3.1.4. The adding value of this study compared with previous studies can be further interpreted and presented in the discussion.

Authors' response:

In response to this comment and R3.1 we have added further discussion to the text focussing on the areas addressed in the reviewer comments.

R3.1.5. The comparisons of estimates for excess deaths between this study and other publications (e.g. <https://jech.bmj.com/content/early/2020/10/15/jech-2020-214764>) can be discussed.

Authors' response:

We have included a comparison of excess death estimates in the discussion (page 8, line 184). See response to R3.1.4.

REVIEWERS' COMMENTS

Reviewer #2 (Remarks to the Author):

The authors addressed all my concerns raised in the first round of reviews. I congratulate them for an interesting and relevant paper.

Reviewer #3 (Remarks to the Author):

Thanks. My comments have been addressed. In your future research, it will be very interesting to see how well the model and factors used in this study can explain the variations in excess mortality across different covid waves in England.